# The Development of a Public Bathroom Perception Scale

**DOI:** 10.3390/ijerph17217817

**Published:** 2020-10-26

**Authors:** Guido Corradi, Eduardo Garcia-Garzon, Juan Ramón Barrada

**Affiliations:** 1Departamento de Psicología, Facultad Salud, Universidad Camilo José Cela, 28692 Madrid, Spain; egarcia@ucjc.edu; 2Departamento de Psicología y Sociología, Universidad de Zaragoza, 44003 Teruel, Spain; barrada@unizar.es

**Keywords:** health psychology, environment evaluation, psychometrics, scale development, well-being, bathrooms, inflammatory bowel diseases

## Abstract

Public bathrooms are sensible locations in which individuals confront an intimate environment outside the comfort of their own home. The assessment of public bathrooms is especially problematic for people whose illnesses make them more prone to needing this service. Unfortunately, there is a lack in the evaluation of the elements that are relevant to the user’s perspective. For that reason, we propose a new scale to assess these elements of evaluation of public bathrooms. We developed a scale of 14 items and three domains: privacy, ease of use and cleanliness. We tested the factor validity of this three-factor solution (*n* = 654) on a sample of healthy individuals and 155 respondents with a bowel illness or other affection that reported to be bathroom-dependent. We found that bathroom-dependent people value more privacy and cleanliness more than their healthy counterparts. We additionally found a gender effect on the scale: female participants scored higher in every domain. This study provides the first scale to assess value concerning public bathrooms and to highlight the relevance of different bathrooms’ aspects to users.

## 1. Introduction

The simple act of evacuating is subject to multiple cultural taboos [1] that obstruct a comprehensive vision of a fundamental human activity. To a greater or lesser extent, people from Western industrialized countries often need to use toilets outside the comfort of their own home, that is, public bathrooms [2]. We have a neglected relation with public toilets, and often the user experience of a public bathroom is not considered. Public bathrooms are ubiquitous in our society (placed in shops, educational centers, offices, etc.). Unfortunately, the use of these bathrooms gives rise to negative psychological disposition and behavior derived from the usability and experience with them [3]. Such a situation may cause emotional consequences that undermine well-being [4,5]. The emotions, fears and thoughts about the use of public bathrooms have been associated with the use of a very intimate place without the usual control and safety offered by the home environment [6,7,8,9]. In other words, using public toilets is, generally, a negative experience with relevant psychological repercussions [10]. Even though different elements in the public bathroom experience could contribute to the construction of these experiences, the study of these elements is limited. To date, the factors that give rise to a valuable public bathroom are missing in the literature. As such, this gap in the evaluation of public bathrooms may well hinder their design, negatively affecting the experience of large sections of the population that are daily users of public bathrooms. In this study, we provide a scale with the elements to evaluate the public bathrooms experience focusing on what people value about them. The proposed scale may unfold distinguishable factors related to public bathroom evaluation. These advances may help to push the study of public bathrooms, a neglected environmental problem. 

From a historical point of view, bathroom conceptions have changed through the centuries [11,12]. Modern bathrooms appeared in the 19th century as merely functional places in the home, intended to satisfy the basic human needs of urinating, defecating and cleaning up afterwards. As new concepts of hygiene arose in Western society, the bathroom became a place where hygiene and cleanliness became the central focus. Nowadays, bathrooms have gone beyond functionality to become a place associated with well-being. Unfortunately, public bathrooms have yet to achieve this sense of well-being and may even trigger feelings of aversion [4,5,8]. In extreme cases, these psychological reactions could become pathological [13,14]. However, there is very little scientific literature about the relationship of people with public bathrooms from a psychological perspective based on their appraisals. The development of these perspectives on assessment may help to improve well-being in different indoor facilities, as some authors plead [15,16]. Further, knowing the axis of evaluation is the first step to understand the raising of these valuations.

Different frameworks like service quality and product evaluation [17,18,19] provide useful tools to research the perceptions about public bathroom use. From the service quality literature, we consider relevant the focus on expectations about the service quality. From the product evaluation where we evaluate appropriately to focus on sensory evaluation aspects and the meeting of functional criteria [20,21]. What both have in common is the relevance in the fact that subjective users’ expectations are a good source of information for product and service evaluation. 

Public bathroom evaluation may differ between groups of a population with different sensitivities and experiences with them. Some people are affected by medical conditions and physical circumstances that lead to a greater and varying need for the use of public bathrooms. These conditions are numerous and include fecal and urinary incontinence, which is known to worsen health and quality of life [4,5]. Further, there are conditions like inflammatory bowel diseases that cause diarrhea [22,23] and affect two million people in Europe [24] and six to eight million worldwide [25]. These diseases are associated with multiple specific psychological effects like irritability, fear and anxiety toward bathroom use [22], feelings of losing personal autonomy [26,27] and particular fears related to toilet urgency [28]. Usually, patients report that bathrooms are valuable for them, and the availability and quality of them have a substantial effect on their lives [29]. While public bathroom use is widespread, there are many potential psychological differences in its usage by patients with inflammatory bowel diseases [30]. 

Moreover, women are a public bathroom-sensitive group as they report concerns on the use of public bathrooms [3]. Some studies report that women’s public bathroom interactions lead to adverse emotional outcomes [3,31]. Regarding the behavior related to public bathrooms, women try to limit the use of public bathrooms as a result of bad experiences with them that may result in bladder and other health problems [32]. Further, built environments tend to be less adjusted to women’s needs in different areas [33,34]. 

In sum, studies on public bathroom use are mostly centered on sanitary aspects [35] or material needs on the safety [36,37] or very concrete elements of usability [38]. So, the relationship between the environment and human perceptions of public bathrooms is a scientific aspect that has not been appropriately explored. Public bathrooms are a built environment that must diminish the shown discomforts related to them, and the first step might be the proper evaluation of their valuable aspects. Finding the relevant elements that drive these evaluations is not evident in literature, so, to develop them, we must rely on different heterogeneous sources. Altogether, we can see that public bathroom use is relevant to people, especially to bathroom-dependent people. Still, there is a gap in the evaluation of what people value specifically about them. Our purpose is to conduct the first endeavor in this area. 

We propose to construct a scale of the perceptions about public bathrooms to measure what people value about them. This tool aims to reveal the elements of evaluation that constitute the perceptions surrounding public bathrooms; that is, the features that give rise to positive or negative assessments of public bathrooms. So, we will focus on which individual elements a participant considers relevant about public bathrooms. Additionally, particular emphasis will be placed on finding whether those elements are similar across the bathroom-dependent and the general population. We hypothesize (H1) that bathroom-dependent participants will consider the different attributes of public bathrooms as more relevant since those participants may be more sensitive to the presence or absence of bathrooms elements. As women face more difficulties when using public bathrooms which are not generally attending their needs [3], we hypothesize gender differences in public bathroom use [39] (H2). We also hypothesized (H3) that frequency of use may be positively correlated with the scale factors as an effect of expertise might make people more attuned to public bathroom features by their expertise [40].

Regarding the avoidance of public bathroom use, we hypothesized (H4) that scale scores would correlate positively with avoiding public bathrooms due to higher awareness of public bathroom features and also the fact that people try not to use them since this use, generally, leads to a bad experience. We further hypothesized that (H5) participants scoring higher in the scale will score higher in their reticence of using public bathrooms due to their awareness of perceived negative features. Another expected relation (H6) is that negative expectations about public bathrooms in some places like bus station and educational center will be positively related to scale scores as awareness of public bathrooms will be linked to examples of public bathrooms. As familiarity with a public bathroom may play a role in expectation formation [31], we will ask participants about familiar and non-familiar bathrooms. 

## 2. Materials and Methods

### 2.1. Scale Development and Item Generation

We developed the items with the help of inflammatory bowel disease patients and validated it with a mixed sample. We obtained qualitative data from hospital visits by the first author of the study. We selected this sample for the development of the item bank because inflammatory bowel disease patients, as it is mentioned before, make constant and urgent use of public bathrooms. This condition turns patients into experts on the use of public bathrooms and the information they give about it is useful to sample the concerns about public bathrooms. We obtained data by means of interviewing a focus group from the Hospital Son Espases (Balearic Islands, Spain) about what constituted a good and a bad experience of a public bathroom. All members of the focus group were patients who were instructed to provide useful information about the elements that give rise to their satisfaction when using a public bathroom [41]. All six participants (three men) were people who came to receive treatment for inflammatory bowel disease at the day-care hospital; all of them reported being a frequent user of public bathrooms. In different sessions, concepts that appeared in these conversations were registered. The annotations were not revised until the end of this part of data collection to not bias participants. We elaborated items based on the comments from their qualitative information. Three main categories were raised when analyzing these comments: privacy, cleanliness and ease of use (see Table 1). Later, categories were double-checked by mapping the words onto the labels and then assigning each comment to a label. Privacy refers to the idea of being in a place where one can be alone and in an environment where one can perform an intimate act [8]. Cleanliness refers to the absence of dirtiness and the presence of a hygienic environment. Ease of use is the label that comprises two close concepts, the environment and its elements of use such as toilet paper and effectiveness, efficiency and satisfaction in the use of that environment [20,42]. We drew up 14 items: 5 for the privacy factor, 5 for the ease of use factor and 4 for the cleanliness factor. These items summarize the valuations proposed by the inflammatory bowel diseases patients. Items are scored with a 5-point scale ranging from 1 = *Totally disagree* to 5 = *Totally agree* to answer how relevant each feature presented is. As an example “*How relevant is to you to be able to close the bathroom door correctly*” for the privacy factor, “*How relevant is to you the bathroom is easy to use*” for ease of use and “*How relevant is to you the bathroom is stain free*” for the cleanliness factor. With these 14 items and three factors, we developed the Public Bathroom Perception Scale (PBPS). The original Spanish items and their English version are available online (https://osf.io/5dnhw/). 

### 2.2. Scale Validation

Data were collected through an online survey. Participants answered the questionnaire, among other study questions reported in the online (https://osf.io/5dnhw/). Answering the study took about 10 min. The first author distributed the link to the survey through social networks like Twitter and Facebook groups, and via email sent to inflammatory bowel disease patient associations. Participants provided informed consent after reading the description, purpose and data management of the study, where the anonymity of the responses was clearly stated. The sample size was not determined a priori; we left open the questionnaire for a reasonable period (one month starting in January 2019). The Ethics Committee from the Universidad Camilo José Cela certified that participants were treated following the Declaration of Helsinki.

Of the respondents, 654 had no missing value (42% male, 0.3% preferred not to respond, one responded “other”, *M*_age_ = 33 years, *SD*_age_ = 9.82). At the same time, 70 participants were excluded due to missing data (data from discarded participants are available in the online repository https://osf.io/5dnhw/). One hundred and fifty-three (23% of the sample) participants responded as having a disease or condition that lead to the frequent use of the bathroom. The bathroom-dependent ill participants indicated to have mostly inflammatory bowel disease (25%), irritable bowel syndrome (25%) and urinary infections (22%). We confirm to report all measures, conditions, data exclusions and how we determined the sample sizes.

### 2.3. Instruments

#### 2.3.1. Sociodemographic

Participants provided information on gender (“*male*”, “*female*”, “*other*” and “*I prefer not to answer*”) and age in years.

#### 2.3.2. Public Bathroom Perception Scale

As previously described, the PBPS is a 14-item scale which aims to assess what people value about public bathrooms with three dimensions: privacy, cleanliness and ease of use, asking about how relevant each feature presented is. The scale responses were provided on a 5-point scale ranging from 1 = *Totally disagree* to 5 = *Totally agree*.

#### 2.3.3. Health Status

Participants informed if they suffer or had ever suffered any disease or condition that led to the frequent use of a bathroom with yes/no as response options and a text box to type their disease. Participants who indicated a positive answer are referred to as bathroom-dependent ill people through the text to handle the heterogeneity of the illnesses that led to this kind of dependence. 

#### 2.3.4. Avoidance and Reluctance to Use Public Bathrooms

Participants indicated the degree to which they, as a rule, avoided the use of public bathrooms. Responses were provided with a 5-point scale ranging from 1 = *Never* to 5 = *Very often*. They also informed how unwilling they were to use public bathrooms in different locations like the workplace, shopping centers, educational centers, bus stations, subway stations, trains, bars and restaurants, commerce and on the street. Responses were provided on a 5-point scale ranging from 1 = *Not at all* to 5 = *Very much*. Reliability was observed to be adequate (ω_ordinal_ = 0.81).

#### 2.3.5. Negative Expectations Relative to Public Bathrooms

Participants indicated their agreement with six statements about public bathrooms’ expectancies: “they are hostile environments”, “they are dirty places”, “there is no privacy”, “they are places where one feels ill at ease”, “they are uncomfortable” and “I would be anxious there”. This set of questions was asked concerning public bathrooms (no additional specification) that the participant was already familiar with such as the office bathroom. Responses were provided on a 5-point scale ranging from 1 = *Totally disagree* to 5 = *Totally agree*. Scores were calculated as the sum of all items as parallel analysis and factor analysis suggest a one-factor solution. Reliability was considered to be sufficient (ω_ordinal Negative expectations_ = 0.88; ω_ordinal Negative expectations familiar bathrooms_ = 0.89).

### 2.4. Data Analyses

#### 2.4.1. Factor Analysis

First, we computed the descriptive statistics of the PBPS items (mean and standard deviation). Secondly, we studied the internal structure of the PBPS with exploratory structural equation modeling (ESEM; [43]). As ESEM, in comparison with exploratory factor analysis (EFA) or confirmatory factor analysis (CFA), is a lesser known technique to assess the internal structure of a questionnaire, we will detail its advantages that justify our selection. For doing so, we will follow previous descriptions [44,45]. 

EFA is usually referred to as a data-driven technique ([46]) and is commonly used with the aim of obtaining a simple and interpretable structure. Basically, and as far as this study is concerned, there are two significant limitations to EFA. First, EFA does not permit the correct evaluation of the measurement invariance across different groups [47]. Measurement invariance implies that the same score has the same interpretation for different groups. The comparability of scores between groups is not something that can be assumed by default, but rather has to be supported by evidence. Further, EFA models cannot be incorporated into a structural model, that is, those latent factors cannot be correlated with additional (latent or observed) variables. Both measurement invariance and structural models can be tested with ESEM models.

CFA is considered a theory-driven technique, as the number of dimensions and the item–factor relationship with which the covariance matrix will be explained must be supported by a strong previous theory or by previous EFAs in which a simple structure has been found. In CFA, the factor loadings are usually estimated with the restriction that each item will only load on the expected factor, the other loadings being fixed to 0. The main limitation of CFA is the restrictive assumption that the factor structure is fully simple [48]. While in the EFA context, simple structure implies no salient loadings on the secondary dimensions, in the CFA context, simple structure means no loading at all. In CFA, any nonmodeled loading different from 0 in the population reduces the model fit and can bias the results. When minor cross-loadings are fixed to 0, the correlations between dimensions are distorted [43,49]. ESEM, like EFA, permits the estimation of the factor loadings of all items in all factors, so that the problem of fixing the cross-loadings to 0 disappears. When the loading matrix of the population includes cross-loadings, ESEM recovers this matrix better than CFA and is not subject to its parameter estimation bias. Given these reasons, we considered ESEM as the best available alternative.

To determine the numbers of dimensions to be retained—which we expected to be three—we used parallel analysis [50] and visual inspection of the scree-plot. Additionally, we considered the theoretical interpretability of the solutions, factor simplicity and factor loading magnitudes, as is recommended in the literature [51,52]. We evaluated the polychoric correlation matrix [53,54]. As an estimator, we used maximum likelihood with robust standard errors and a mean and variance adjusted test statistic (WLSMV in Mplus 8.0 [55]) with geomin rotation. The goodness of fit of all the derived models was assessed with the common cut-off values for the fit indices [56]. CFI and TLI with values greater than 0.95 and RMSEA less than 0.06 were indicative of a satisfactory fit. It should be noted that those cut-offs were developed for confirmatory factor analysis with continuous responses, so those values should be considered with caution. To assess reliability, we computed McDonald’s ordinal omega (ω) [57,58]. 

#### 2.4.2. Differences between Group Scores and Correlations

We investigated whether the PBPS structure was invariant with regard to bathroom dependency and gender. With regard to measurement invariance, we followed traditional guidelines to check whether scalar invariance could be achieved using multigroup ESEM. Thus, we compared unrestricted, configural (a model where similar items are expected to load into the same factors) and scalar models (a model where factor loadings and intercepts are constrained to be equal across groups). Noteworthy, metric invariance cannot be assessed when taking into account categorical variables in ESEM [43]. Due to the categorical nature of the variables, and to avoid computational problems due to the presence of empty cells, items were dichotomized (response equal to *Totally agree* or not) to conduct these analyses. We assessed invariance considering chi-square tests between constrained models (where the most restricted model was retained if a non-significant test is observed [43]).

Additionally, we also explored whether the most constrained model presented a model decrease fit (assessed through CFI, TLI or RMSEA explorations) larger than 0.01 to reject invariance [59,60]. After we established scalar invariance, we compared the standardized latent means of the factors to understand group differences, using dependent individuals and females as the reference group. Lastly, we extended the ESEM model to explore the relationship between the PBPS and other variables of interest. For negative expectations relative to public bathrooms and reticence, we fitted unidimensional measurement models. Afterwards, we compared latent correlations of these factors with PBPS factors. 

### 2.5. Statistical Software

We performed all analyses in Mplus 8.0 (StatModel, Los Angeles, USA) [55], Jamovi (The jamovi Project, Sydney, Australia) [61] and R (R Foundation for Statistical Computing, Vienna, Austria) [62]. For the latter, we used the following packages: *tidyverse* and *ggplot2* [63] for data wrangling and plotting. Data and scripts are provided at https://osf.io/5dnhw/.

## 3. Results

### 3.1. Items Descriptive Analysis

The mean item scores were high, ranging from 3.89 (“*The bathroom is stain-free*”) to 4.77 (“*The bathroom being clean*”) in a 1-to-5 range. The overall mean item score was 4.20. Standard deviations ranged from 0.59 to 1.18 (see Table 1).

### 3.2. Factor Analysis

The parallel analysis suggested two components (see Figure 1), with the third factor almost coincident with the eigenvalues of the random data. So, we fitted models with two and three factors and compared the results (see Table 1). As could be expected, the three-factor model revealed the best performance on different indices and RMSEA (CFI = 0.98, TLI = 0.96, RMSEA = 0.06 95% CI [0.06, 0.08]) when compared with the two-factor solution (CFI = 0.89, TLI = 0.85, RMSEA = 0.15 95% CI [0.14, 0.16]). 

For the two-factor solutions, ease of use items showed a complex structure with meaningful cross-loadings in both factors, hampering the interpretation of both items and factors. We chose the three-factor model as it is the most interpretable structure [50], which followed the expected loadings pattern. The mean primary loading was high for all three factors (*M*_λ_ = 0.71, for privacy, range [0.51, 95]; *M*_λ_ = 0.79, for ease of use, range [0.67, 86]; and *M*_λ_ = 0.74, for cleanliness, range [0.69, 84]). The secondary loadings were rather small, with mean unsigned loadings equal to 0.08. The only item with relevant cross-loading was the “To be able to close the bathroom door correctly”, which loaded moderately in the ease of use factor. Latent correlation between factors was moderate to high: privacy and ease of use showed the highest correlation (*r* = 0.60, 95% CI [0.52, 0.68]), with cleanliness and privacy (*r* = 0.46, 95% CI [0.31, 0.64]), and cleanliness and ease of use being of similar magnitude (*r* = 0.47, 95% CI [0.37, 0.60]). Accordingly, we retained the three-factor solution as our final solution. For this factor structure, factors showed an acceptable reliability: cleanliness (ω = 0.84), privacy (ω = 0.88) and ease of use (ω = 0.91). The total percentage of explained variance was 65.86%.

As can be seen in Figure 2, the following item descriptive results, means and medians for the different dimensions were high: above 4 in all three scale means in the 1-to-5 scoring range. Results showed that cleanliness mean scores were the highest (*M* = 4.28, *Mdn* = 4.50, *SD* = 0.71); meanwhile, ease of use scores were the lowest (*M* = 4.12, *Mdn* = 4.20, *SD* = 0.80) with privacy scores in between (*M* = 4.20, *Mdn* = 4.40, *SD* = 0.80). All three score distributions were negatively skewed.

### 3.3. Difference on Scores by the Presence of Bathroom Dependency and by Gender

Prior to investigating the differences between bathroom dependency (bathroom-dependent: *n* = 155; non-dependent: *n* = 499) and gender groups (female: *n* = 378; male: *n* = 272), we aimed to establish PBSP measurement invariance. For the latter group, we excluded the participants labeled as “other” due to their sample size (*n* = 5). Results showed that the PBPS achieved this goal with regard to both groups (Table 2) as neither significant chi-square tests nor decreases above 0.01 in CFI or TLI were observed. Thus, comparisons between means of the group ensued. 

The comparison of latent means for the scalar models evidenced that female participants valued all factors to a higher degree than male participants. In detail, females presented higher scores on privacy (difference = −0.32, *SE* = 0.08; *p* < 0.001), ease of use score (difference = −0.40, *SE* = 0.17; *p* = 0.018) and cleanliness (difference = −0.54, *SE* = 0.12; *p* < 0.001). Regarding health status, we observed that bathroom-dependent participants valued privacy (difference = 0.27, *SE* = 0.10; *p* < 0.001) and cleanliness (difference = 0.02, *SE* = 0.08; *p* = 0.035) to a higher degree than non-dependent participants. However, dependent and non-dependent participants valued the ease of use of bathrooms to a similar degree (difference = 0.07; *SE* = 0.12; *p* = 0.546). 

### 3.4. Correlations between Variables

We investigated the external validity of each of the PBPS scores by expanding our previous ESEM model to explore the latent correlation between the PBPS and a set of meaningful, related variables (Table 3). This model presented an overall fit slightly worse than the proposed cut-offs (*Χ*^2^(581) = 2625.34; *p* < 0.001; CFI = 0.92; TLI = 0.90; RMSEA = 0.07, 95%, CI [0.071, 0.076]). The result evidenced that as a general pattern, cleanliness and privacy scales showed higher correlations with external variables than ease of use. Non-significant correlations were found between frequency of use and each factor mean scores (*r* = 0.02, *r* = −0.03, and *r* = −0.09), with the highest between the negative expectations and scale scores (*r* = 0.28, *r* = 0.15, and *r* = 0.28). Noteworthy, ease of use scores were not correlated with negative expectations to use public and familiar bathrooms. 

## 4. Discussion

In this study, we present a new perception of the Public Bathroom Perception Scale (PBPS). This tool consists of 14 items to assess the participants’ evaluations about the use of public bathrooms. The presented scale is divided into three domains: privacy, ease of use and cleanliness. These domains depict the elements that participants value about public bathrooms. We tested the proposed scale on a Spanish sample. In our proposed scale, we report critical correlations as external validity, and we point out the relevance of public bathrooms for bathroom-needing ill people. 

A relevant aspect of this investigation was the comparison of bathroom-dependent ill people and a sane counterpart in the PBPS scale scores. We showed how the mean scores of latent punctuations (H1) in privacy and cleanliness were significantly higher in participants who indicated to have any illness that makes the use of public bathrooms more frequent. The different use of ill participants could explain this result. They are more sensitive towards public bathroom use due to their illness; they find these measured scales more valuable and relevant. We did not find any differences regarding the ease of use scores between these groups. One tentative interpretation is that bathroom-dependent people are not differentially sensitive to the easiness of use. An alternative interpretation could be found in the users’ difficulties to correctly identify usability aspects of the ease of use when assessing the architectural environments without experiencing the assessed object in real time and the relatively low salience of this feature in architectural environments [64,65,66,67]. This result highlights the fact that bathroom-dependent ill people value some of these aspects of public bathrooms more than their sane counterparts, at least in the privacy and cleanliness factors. 

Analysis by gender revealed differences in each factor: female participants systematically scored higher in the scale scores (H2). This gender difference evidences how female participants value each attribute of a public bathroom more than their male counterparts. This difference in the valuations may be the effect of differences in the use of public bathrooms and the way the public bathrooms are built and maintained [33,34]. Built environments are generally evaluated as less satisfying by women [68] due to design which fails to take into account the different and specific needs [69]. Even the vandalism on public bathrooms shows gender differences [70]. These differences in scale scores may affect some of the reported problematic uses of bathrooms by women [32].

We fund some behavioral consequences of the concern around public bathrooms. Contrary to our expectations, the frequency of use was not related to any score on the scale (H3). Avoidance of using public bathrooms was related to higher scores on all factor scales (H4), especially the cleanliness and privacy factors. Further, results identified a positive relation between reluctance to use public bathrooms, and all factor scores meaning that participants who are more concerned about these factors (H5) are also more reticent to use public bathrooms, but the relation is small. Altogether, these results show that people more aware of concerns about public bathrooms do not try to avoid the use of them, but they do not want to use them. The lack of an effect of experience might be responsible for the absence of substantive correlations between factor scores and the frequency of use. We expected that the greater the public bathroom use, the larger the concern around the elements involved, but this effect was not found. However, we found that people who value the different aspects of public bathrooms present on the scale do not avoid, but are reticent to use, them.

As previously stated, one key point of the evaluation of services is the expectations about the product [71,72,73]. We hypothesized that negative expectations about examples of public bathrooms would be positively related to scale scores (H6). We asked participants about their expectations about public bathrooms and familiar public bathrooms, finding noteworthy relations with the different PBPS factors. Negative expectations are more related to cleanliness and privacy and to a lesser degree to ease of use. When the same questions are asked about familiar public bathrooms, the relationships are lower than the non-familiar ones, and follow a similar pattern: the privacy factor is related to a lower degree than cleanliness, but ease of use is not statistically different. These results highlight the relevance of expectations. Users’ negative expectations of specific categories of public bathroom use rise depending on how relevant they consider these aspects to be. The lesser relations between the familiar and unfamiliar public bathrooms highlight the role of familiarity [9] in bathroom perceptions.

We showed different elements that constitute a public bathroom valuable: privacy, ease of use and cleanliness. These elements might guide the design of this kind of constructed environment when the users’ perspective is considered to improve well-being [74,75]. The scale scores may help the decision-makers and researchers interested in the detection of differences between populations regarding the bathroom valuations. This study is the first to characterize different factors that guide the user’s valuations and may be the first step to inquire about the determinants in the public bathroom negative evaluations. Knowing about the differences in public bathroom valuations could help the understanding of the patients’ quality of life disturbances related to public bathroom use. Further, knowing the relations between users’ valuations of public bathrooms with other aspects like the reticence to use them or the negative expectations towards them may encourage research on the topic and uncover business opportunities.

In this study, we illustrated a newly developed scale and checked differences between scores of bathroom-dependent ill people and sane counterparts. However, there are more areas in which relevant differences might occur and should be explored in future research. One unexplored topic is the differences in bathroom use by different communities [39,76]. Future research may focus on the psychometric properties of the inventories used in this study which may lead to different correlations due to measurement error. Moreover, future research might handle the specific consequences on behavior due to the other stated negative emotions that public bathrooms elicit and the issues associated with them. Another critical point future research should investigate is the difference in personal relevance of each scale: the degree to which privacy may be more relevant to ease of use, and cleanliness more relevant to that first variable, needs understanding. Ecological validity of the results may be incremented using virtual reality technology or real stimuli to elicit reactions and take a more valid assessment [77] and also overcome the bias from the non-random sampling when testing may be a future issue. 

Taken together, the present study is relevant to public policy design, evaluation and environmental planning. As we showed, public bathrooms use is an appropriate topic for people that, related to negative expectations, show reticence to use but do not to avoid the use of them. Through the evaluation of constructed environments with some simple items, designers might improve the quality of the offered products and satisfy a fundamental human need, ensuring a good experience, and also provide an integrative health response to different chronically ill patients [78]. Our proposed scale may enhance the design of public bathrooms to build better environments focusing more on the user’s perspective and needs. 

## 5. Conclusions

In this study, we present the first scale for assessing the relevance of the privacy, ease of use and cleanliness of public bathrooms, made focusing on users. We introduced a new tool for evaluating public bathrooms with implications for public health policy and quality evaluation with a scale that shows good internal validity and correlations with behavior and users’ expectations. The three different axes of relevance to public bathroom users have implications for well-being as public bathrooms play a neglected but relevant role in people’s lives.

## Figures and Tables

**Figure 1 ijerph-17-07817-f001:**
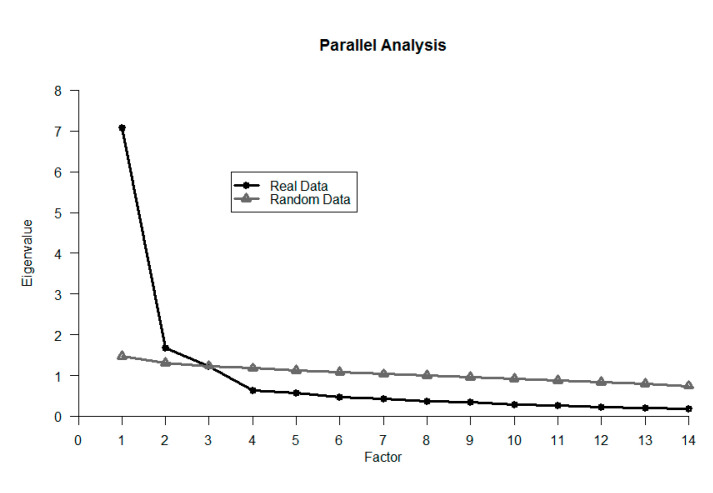
Parallel analysis of the Public Bathroom Perception Scale.

**Figure 2 ijerph-17-07817-f002:**
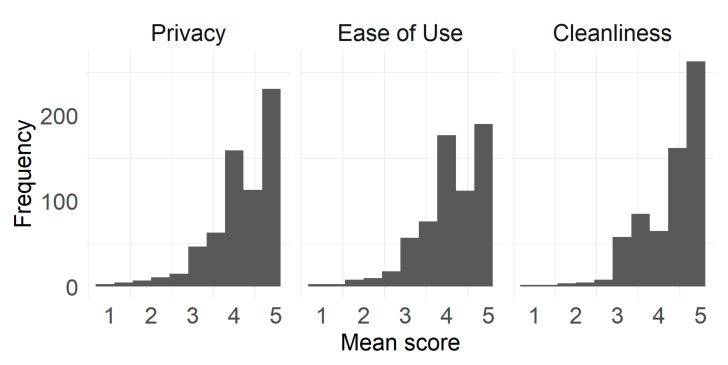
Distribution of mean scores of each factor.

**Table 1 ijerph-17-07817-t001:** Items, mean score and item loadings of the Public Bathroom Perception Scale (PBPS) for three- and two-factor solutions.

Dimension/Item	Descriptive Statistics		Loadings Three-Factor Solution			Loadings Two-Factor Solution	
	M	*SD*	PR	EU	CL	F1	F2
Privacy							
The privacy it offers	4.18	1.05	**0.95**	−0.02	−0.09	**0.96**	−0.22
The bathroom being isolated from the rest of the place	3.92	1.16	**0.51**	0.24	−0.02	**0.65**	0.00
To be able to close the bathroom door correctly	4.58	0.80	**0.53**	**0.38**	0.05	**0.76**	0.12
The privacy offered by the bathroom	3.98	1.18	**0.90**	−0.01	−0.14	**0.95**	−0.24
The bathroom not feeling exposed in the bathroom	4.37	1.00	**0.64**	0.17	0.04	**0.74**	0.01
Ease of use							
The bathroom had everything on hand when needed	4.14	0.96	0.08	**0.79**	−0.01	**0.61**	**0.31**
The bathroom had everything needed	4.32	0.88	0.09	**0.79**	−0.01	**0.62**	**0.31**
The bathroom is easy to use	3.92	1.07	−0.01	**0.86**	−0.05	**0.57**	**0.31**
The bathroom not showing uncomfortable traits when using it	4.13	0.96	0.12	**0.67**	0.07	**0.56**	**0.32**
The bathroom is functional (thought for easy and fast use)	4.07	1.02	−0.05	**0.85**	0.07	**0.54**	**0.44**
Cleanliness							
The bathroom being clean	4.77	0.59	0.01	0.20	**0.74**	0.09	**0.79**
The bathroom was showing neutral odor.	4.30	0.88	−0.08	0.19	**0.69**	0.01	**0.74**
The bathroom showing no signs of use	3.86	1.16	0.06	−0.02	**0.69**	0.01	**0.66**
The bathroom is stain-free	4.20	1.00	0.04	0.00	**0.84**	−0.02	**0.81**

PR = privacy, EU = ease of use, CL = cleanliness. Items from the PBPS, all items were asked with a preceding “On a public bathroom how relevant is to you…”. All loadings above 0.30 are shown in bold. Shaded cells represent the expected pattern of loadings.

**Table 2 ijerph-17-07817-t002:** Multigroup invariance exploratory structural equation modeling (ESEM) for the PBPS and gender and bathroom dependency groups.

Model	Χ^2^ (*df*)	ΔΧ^2^ (*df*)	CFI	TLI	RMSEA
*Gender*					
Configural	116.97 (104)	-	0.998	0.997	0.02 [0.00, 0.04]
Scalar	140.46 (134)	27.79(30)	0.999	0.999	0.01 [0.00, 0.03]
*Bathroom dependency* ^†^					
Configural	155.05 (106) *	-	0.994	0.989	0.04 [0.03, 0.05]
Scalar	162.03 (76)	22.91(30)	0.996	0.995	0.03 [0.00; 0.04]

* significant at 0.001 level. ^†^ A Heywood case was detected for item 1 (i.e., “The privacy it offers”) in both models. Χ^2^ (*df*) = chi-square statistic with degrees of freedom in parenthesis. ΔΧ^2^ (*df*) = chi-square difference. CFI = comparative fit index; TLI = Tucker–Lewis index. RMSEA = root mean square error of approximation (with 95% confidence interval in brackets).

**Table 3 ijerph-17-07817-t003:** Descriptive statistics of variables and latent correlations between variables and scores.

Variable	Privacy	Ease of Use	Cleanliness	*M*	*SD*	Range
Frequency of use	0.02 [−0.06, 0.10]	−0.03 [−0.11, 0.05]	−0.09 [−0.19, 0.01]	3.58	0.98	[1, 5]
Use avoidance	**0.20 [0.12, 0.28]**	**0.09 [0.01, 0.17]**	**0.24 [0.15, 0.32]**	3.08	1.15	[1, 5]
Negative expectation	**0.28 [0.23, 0.33]**	**0.15 [0.10, 0.21]**	**0.28 [0.22, 0.34]**	19.97	5.2	[7, 23]
Negative expectation (familiar bathrooms)	**0.23 [0.17, 0.29]**	0.05 [−0.01, 0.11]	**0.17 [0.10, 0.24]**	13.57	5.18	[6, 30]
Reticence	**0.05 [0.02, 0.21]**	**0.03 [0.01, 0.06]**	**0.07 [0.04, 0.10]**	25.29	6.97	[9, 45]

Values show latent correlations. Brackets show the 95% confidence interval, bold numbers depict correlations with *p*-values < 0.05.

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
