# Peer review of "The Development of a Public Bathroom Perception Scale"

_ijerph, 2020, doi:10.3390/ijerph17217817_

Round 1

Reviewer 1 Report

I have confirmed the response letter from the authors but felt that some critical issues were left. For example, it was not easy to understand the significance of this study, which was pointed out in my first review. Therefore, I will recommend "reject" even if I review the paper again.

Author Response

We acknowledge the comment, but we think that the manuscript it has significantly improved and the concerns adressed. We added examples of potential use and a significance statement. Also, we want to point that research on the relation between people and public bathrooms is at it beginning. So, it's hard to predict its promise and future relevance. In our study we characterize a new construct, thus is, public bathroom valuation and we show how it is related to different outcomes like the reticence to use them. Potentially, our scale may help future researchers interested in explaining which experiences give rise to these different valuations. As example, future researchers might use the our scale to track changes in the relevance placed to the different axis uncovered by the our study when applying some intervention to the environment. Also, it can be extended to other populations of interest. Our group is working in the quality of life disturbances caused by the struggles of poor quality public bathroom. We are using the PBPS to assess the relevance of each axis to the different outcomes related to well-being and find in which degree the assigned relevance of each axis drives to specific losses on patients´ quality of life. As we find significant differences between valuations made by women and men, we also aim to study the environmental differences which give raise to these differences. As our preliminary results caught the attention of different institutions, patients associations and foundations dedicated to well-being, we think it have impredictable relevance and we hope that it may push the research towards a neglected environmental problem like the quality of public bathrooms.

Especifically, we updated the following sections:

On in 1 .Introduction:

The proposed scale may unfold disntinguishible factors related to public bathroom evaluation. These advances may help to push on the study of public bathrooms, a neglected environmental problem.

And in the 4. Discussion

We show different elements that constitute a public bathroom valuable: Privacy, Ease of use and Cleanliness. These elements might guide the design of this kind of constructed environments when user's perspective is considered to improve wellbeing [77,78]. The scale scores may help the decision makers and researchers interested in the detection of differences between populations regarding the bathrooms valuations. This study is the first to characterize different factors that guide user’s valuations and may be the first step to the inquire in the determinants on the public bathroom negative evaluations. Knowing about the differences in public bathroom valuations may help the understanding of the patients' quality of life disturbances related to public bathroom use. Also, knowing the relations between users valuations of public bathrooms with other aspects like the reticence to use them or the negative expectations towards them may encourage the research on the topic and uncover business opportunities.

Reviewer 2 Report

The authors have done a very good job in revising the manuscript. However, I do have noticed a few issues that need clarification.

(1)  page 2, lines 54-55: Please double check; it seems that information is missing.

(2)  section 2.4.2 and section 3.3 – group differences: In order to compare the means of different groups, measurement invariance of the scale has to be ensured (e.g., Putnick & Bornstein, 2016). The present results and conclusions can therefore not (yet) be interpreted/evaluated.

(3)  table 2: As measurement error was already mentioned as a limitation of the present study, why not using latent correlations instead of Spearman correlations?

(4)  Online supplement

a.    normality_and_reliability.pdf: It is unclear which scale belongs to which result (e.g., reliability analysis)

b.    Spanish_item_list.docx: It might be advantageous to explicitly link the Spanish items with the English items (Table 1) by providing the item names in Table 1.

*References*

Putnick, D. L., & Bornstein, M. H. (2016). Measurement invariance conventions and reporting: The state of the art and future directions for psychological research. Developmental Review, 41, 71–90. https://doi.org/10.1016/j.dr.2016.06.004

Author Response

The authors have done a very good job in revising the manuscript. However, I do have noticed a few issues that need clarification.

We thank the reviewer for the precise comments.Taking into account reviewer's comments, we have changed our analytical approach. We have switched from EFA to exploratory structural equation modeling (ESEM). By doing so, we can keep some of the advantages of our previous EFA approach (fit indices are available for EFA and ESEM techniques, no distorsion of the estimated parameters due to fixing to 0 cross-loadings), but we can now incorporate measurement invariances tests and the ESEM factors to a structural model. As we anticipate that the ESEM technique may not be well known by potential readers of the manuscript, we have detailed its characteristics and advantages:

“First, we computed the descriptive statistics of the PBPS items (mean and standard deviation). Secondly, we studied the internal structure of the PBPS with exploratory structural equation modeling (ESEM; [1]). As ESEM, in comparison with exploratory factor analysis (EFA) or confirmatory factor analysis (CFA), is a lesser known technique to assess the internal structure of a questionnaire we will detail its advantages that justify our selection. For doing so, we will follow previous descriptions[1,6]. 

EFA is usually referred to as a data-driven technique ([2]) and is commonly used with the aim of obtaining a simple and interpretable structure. Basically, and as far as this study is concerned, there are two important limitations to EFA. First, EFA does not permit the correct evaluation of the measurement invariance across different groups [3]). Measurement invariance implies that the same score has the same interpretation for the different groups. The comparability of scores between groups is not something that can be assumed by default, but rather has to be supported by evidence. Also, EFA models cannot be incorporated into a structural model, that is, those latent factors cannot be correlated with additional (latent or observed) variables. Both measurement invariance and structural models can be tested with ESEM models.

CFA is considered a theory-driven technique, as the number of dimensions and the item-factor relationship with which the covariance matrix will be explained must be supported by a strong previous theory or by previous EFAs in which a simple structure has been found. In a CFA, the factor loadings are usually estimated with the restriction that each item will only load on the expected factor, the other loadings being fixed to 0. The main limitation of CFA is the restrictive assumption: The factor structure is fully simple [4]. Whereas in the EFA context, simple structure implies no salient loadings on the secondary dimensions, in the CFA context, simple structure means no loading at all. In CFA, any nonmodeled loading different from 0 in the population reduces the model fit and can bias the results. When minor cross-loadings are fixed to 0, the correlations between dimensions are distorted [1,7]. ESEM, like EFA, permits the estimation of the factor loadings of all items in all factors, so that the problem of fixing the crossloadings to 0 disappears. When the loading matrix of the population includes cross-loadings, ESEM recovers this matrix better than CFA and is not subject to its parameter estimation bias. Given these reasons, we considered the ESEM as the best available alternative.”

(1) page 2, lines 54-55: Please double check; it seems that information is missing.

We thank the comment and we rephrased the paragraph. The paragraph is updated to:

Different frameworks like service quality and product evaluation [17–19] provide useful tools to research the perceptions about public bathroom use. From the service quality literature, we consider relevant the focus on expectations about the service quality and form the product evaluation we consider relevant the focus on sensory evaluation aspects and the meeting of functional criteria [20,21]. What both have in common is the relevance that the fact that subjective users' expectations are a good source of information for product and service evaluation.

(2) section 2.4.2 and section 3.3 – group differences: In order to compare the means of different groups, measurement invariance of the scale has to be ensured (e.g., Putnick & Bornstein, 2016). The present results and conclusions can therefore not (yet) be interpreted/evaluated.

We thank the reviewer suggestion and we performed the measurement invariance study. We added it to the manuscript. We have justified why those tests were needed in the Factor analysis section.

Measurement invariance implies that the same score has the same interpretation for the different groups. The comparability of scores between groups is not something that can be assumed by default, but rather has to be supported by evidence.

We have explained how those tests were carried out in the Differences between group scores and correlations section:

We investigated whether PBPS structure was invariant with regards to bathroom dependency and gender. With regards to measurement invariance, we followed traditional guidelines to check whether scalar invariance could be achieved using multigroup ESEM. Thus, we compared unrestricted, configural (a model where similar items are expected to load into the same factors) and scalar models (a model where factor loadings and intercepts are constrained to be equal across groups). Noteworthy, metric invariance cannot be assessed when taking into account categorical variables in ESEM [55]. Due to the categorical nature of the variables, and to avoid computational problems due to the presence of empty cells, items were dichotomized (response equal to Totally agree or not) to conduct these analyses. We assessed invariance considering chi-square tests between constrained models (where the most restricted model was retained if a non-significant test is observed [55]).

Additionally, we also explored whether the most constrained model presented a model decrease fit (assessed through CFI, TLI or RMSEA explorations) larger of .01 so to reject invariance. After we established scalar invariance, we compared the standardized latent means of the factors so to understand group differences, using dependent individuals and females as the reference group.

And these are the reported results (Difference on scores by the presence of bathroom dependency and by gender section):

Prior to investigate differences between bathroom dependency (bathroom dependent: n = 155; non-dependent: n = 499) and gender groups (female: n = 378; male: n = 272), we aimed to establish PBSP measurement invariance. For the latter group, we excluded the participants labelled as "other" due to their sample size (n = 5). Results showed that PBPS achieved this goal with regards to both groups (Table 2) as neither significant chi-square tests nor decreases above .01 in CFI or TLI were observed. Thus, comparisons between means of the group ensued.

--- Table 2 ---

The comparison of latent means for the scalar models evidenced that female participants valued all factors to a higher degree than male participants. In detail, females presented higher scores on Privacy (difference = –.32, SE = .08; p <.001), Ease of Use score (difference = –.40, SE = .17; p =.018), and Cleanliess (difference = –.54, SE = .12; p < .001). Regarding health status, we observed that bathroom dependent participants valued Privacy (difference = .27, SE = .10; p < .001) and Clealiness (difference = .02, SE = .08; p = .035) to a higher degree than non-depedent participants. However, dependent and non-depedent participants valued the Ease of Use of bathrooms to a similar degree (difference =.07; SE =.12; p = .546) .

(3) table 2: As measurement error was already mentioned as a limitation of the present study, why not using latent correlations instead of Spearman correlations?

A benefit from moving to an ESEM framework was possibility of including our criterion variable within a structural model and to estimate latent correlation between PBPS scores and those variables.

We have explained what we have done:

Lastly, we extended the ESEM model to explore the relationship between PBPS and other variables of interest. For negative expectations relative to public bathrooms and reticence, we fitted unidimensional measurement models. Afterwards, we compared latent correlations of these factors with PBPS factors.

And we have described our results for this structural model:

We investigated the external validity of each of the PBPS scores by expanding our previous ESEM model to explore the latent correlation between PBPS and a set of meaningful, related variables (Table 3). This model presented a overall fit slightly worse than the proposed cut-offs (Χ2(581) = 2625.34; p <.001; CFI = .92; TLI = .90; RMSEA = .07, 95%CI [.071,.076]). Result evidenced that as a general pattern, Cleanliness and Privacy scale showed higher correlations with external variables than Ease of Use. Non-significant correlations were found between frequency of use and each factor mean scores (r = .02, r = –.03, and r = –.09), with the highest between the negative expectations and scale scores (r = .28, r = .15, and r = .28). Noteworthy, Ease of Use scores was not correlated to negative expectations to use public and familiar bathrooms.

--- Table 3 ---

(4) Online supplement

  1. normality_and_reliability.pdf: It is unclear which scale belongs to which result (e.g., reliability analysis)

We added clarification to the document.

  1. Spanish_item_list.docx: It might be advantageous to explicitly link the Spanish items with the English items (Table 1) by providing the item names in Table 1.

We added the English translation to the document.

References

  1. Asparouhov, T.; Muthén, B. Exploratory structural equation modeling; 2009; Vol. 16; ISBN 1070551090.
  2. Fabrigar, L.R.; Maccallum, R.C.; Wegener, D.T.; Strahan, E.J. Evaluating the use of exploratory factor analysis in psychological research. Psychol. Methods 1999, 4, 272–299, doi:10.1037/1082-989X.4.3.272.
  3. Meredith, W. Measurement invariance, factor analysis and factorial invariance. Psychometrika 1993, 58, 525–543, doi:10.1007/BF02294825.
  4. Marsh, H.W.; Morin, A.J.S.; Parker, P.D.; Kaur, G. Exploratory structural equation modeling: An integration of the best features of exploratory and confirmatory factor analysis. Annu. Rev. Clin. Psychol. 2014, 10, 85–110, doi:10.1146/annurev-clinpsy-032813-153700.
  5. Garrido, L.E.; Barrada, J.R.; Aguasvivas, J.A.; Martínez-Molina, A.; Arias, V.B.; Golino, H.F.; Legaz, E.; Ferrís, G.; Rojo-Moreno, L. Is Small Still Beautiful for the Strengths and Difficulties Questionnaire? Novel Findings Using Exploratory Structural Equation Modeling. Assessment 2020, 27, 1349–1367, doi:10.1177/1073191118780461.
  6. Ramos-Villagrasa, P.J.; Barrada, J.R.; Fernandez-del-Rio, E.; Koopmans, L. Assessing Job Performance Using Brief Self-report Scales : The Case of the. J. Work Organ. Psychol. 2019, 35, 195–205.
  7. Garcia-Garzon, E.; Abad, F.J.; Garrido, L.E. On Omega Hierarchical Estimation: A Comparison of Exploratory Bi-Factor Analysis Algorithms. Multivariate Behav. Res. 2020, 0, 1–19, doi:10.1080/00273171.2020.1736977.

Round 2

Reviewer 2 Report

Well done!

This manuscript is a resubmission of an earlier submission. The following is a list of the peer review reports and author responses from that submission.

Round 1

Reviewer 1 Report

This study developed a novel scale related to perceptions of the use of public bathrooms and showed the validity and reliability of the scale and the difference of the score between people with and without disability.

  1. It is unclear the importance of the scale; that is, it is difficult to understand the significance of the scale. How is the scale effect for other researchers or policymakers? I think it is obvious that privacy, ease of use, and cleanliness are important in terms of the use of a public bathroom. Why is it needed to develop a new scale for assessing these points?

  1. This study showed that the characteristics in perception for the use of public bathrooms are different between people with and without disease. However, that may be obvious because the PBPS was developed based on the qualitative data obtained from the six participants with inflammatory bowel disease. In other words, the PBPS may be strongly reflected in perceptions of disease people because the data extracted from healthy people are not included to develop the scale. The authors are needed to add rational reasons to explain the validity of the process of development of the PBPS.

  1. It is better to show more information about data collection. The authors used social networks in this study. Was the link distribution carried out randomly? Do the authors think that there is a sampling bias?

  1. The presence of ill is a rough explanation. It is thought that the health status of participants who are defined as non-healthy people in this study was not common.

  1. Is there a gender difference in the scale? Investigating gender differences may lead to expanding the usability of the scale.

Reviewer 2 Report

The manuscript describes the development, psychometric evaluation and initial validation of the Public Bathroom Perception Scale (PBDS). In general, it is an interesting (and somewhat unusual) topic, the statistical analyses are appropriate, and the manuscript is well written. I enjoyed reading the manuscript! However, I do have some minor issues and recommendations, which might help to improve the manuscript.

(1) line 145: Was there a specific reason why the 65 participants with missing data were excluded? Listwise deletion is considered the worst solution in the literature regarding missing data. Multiple imputation (MI) or Full Information Maximum Likelihood (FIML) is usually considered as the better alternative.

(2) line 188: The requirements for Cronbach's alpha are not met in this study (see Table 1). The authors should use an alternative such as McDonald's omega (e.g., Dunn et al., 2013)

(3) line 205: The data and scripts are not freely available at OSF (i.e., you have to request permission to access the data).

(4) Table 1 / line 239: Item and scale scores indicate ceiling effects and variance restrictions, which might have implications for the further analyses (e.g., small correlations between the scales and other variables). This issue should be considered and discussed.

(5) lines 232-233: The correlation between Ease of Use and Privacy is relatively high (r = .61). Gignac et al.(2017) showed that even such moderate correlations do not guarantee distinct factors. They proposed an additional approach based on the Schmid-Leiman transformation and Omega hierarchical subscale to further investigate the factor structure. It might be useful to apply this procedure here, too.

(6) line 254: I don´t see why Fig. 1 is related to the mean score of each participant.

(7) paragraph 3.4 / Table 2:

- In order to evaluate the correlations between the variables (by the way: this is not a meaningful heading for the section), it is necessary to present the descriptive statistics for the other variables, too (see comment above regarding ceiling effect).

- It should be investigated (or at least mentioned) whether a sum score is an appropriate solution for the other scales (e.g. unidimensionality based on factor analyses). The low correlations could also be a consequence of poor psychometric quality of the other scales.

- It is not necessary to provide different asterisks and statistical significance levels: Either the result at a previously defined level (e.g., alpha = .05) is significant or not (e.g., Lakens et al., 2018).  Another way would be to abandon it altogether and present confidence intervals instead (see Cumming, 2013).

(8) line 324: Gender was included in the survey. It is not clear why this question was not examined in the present study.

Final statement

I request that the authors add a statement to the paper confirming whether, for all experiments, they have reported all measures, conditions, data exclusions, and how they determined their sample sizes. The authors should, of course, add any additional text to ensure the statement is accurate. This is the standard reviewer disclosure request endorsed by the Center for Open Science [see http://osf.io/hadz3]. I include it in every review.

*References*

Cumming, G. (2013). The New Statistics: Why and How. Psychological Science, 25(1), 7–29. ja. https://doi.org/10.1177/0956797613504966

Dunn, T. J., Baguley, T., & Brunsden, V. (2013). From alpha to omega: A practical solution to the pervasive problem of internal consistency estimation. British Journal of Psychology, 1–14. https://doi.org/10.1111/bjop.12046

Gignac, G. E., & Kretzschmar, A. (2017). Evaluating dimensional distinctness with correlated-factor models: Limitations and suggestions. Intelligence, 62, 138–147. https://doi.org/10.1016/j.intell.2017.04.001

Lakens, D., Adolfi, F. G., Albers, C. J., Anvari, F., Apps, M. A. J., Argamon, S. E., Baguley, T., Becker, R. B., Benning, S. D., Bradford, D. E., Buchanan, E. M., Caldwell, A. R., Van Calster, B., Carlsson, R., Chen, S.-C., Chung, B., Colling, L. J., Collins, G. S., Crook, Z., … Zwaan, R. A. (2018). Justify your alpha. Nature Human Behaviour, 2(3), 168–171. https://doi.org/10.1038/s41562-018-0311-x